# Enteral Nutrition in Adult Crohn’s Disease: Toward a Paradigm Shift

**DOI:** 10.3390/nu11092222

**Published:** 2019-09-14

**Authors:** Simona Di Caro, Konstantinos C. Fragkos, Katie Keetarut, Hui Fen Koo, Gregory Sebepos-Rogers, Hajeena Saravanapavan, John Barragry, Jennifer Rogers, Shameer J. Mehta, Farooq Rahman

**Affiliations:** 1Intestinal Failure Service, GI Services, University College London Hospitals NHS Foundation Trust, 250 Euston Road, London NW1 2PG, UK; simona.dicaro@nhs.net (S.D.C.); constantinos.frangos.09@ucl.ac.uk (K.C.F.); g.sebepos-rogers@nhs.net (G.S.-R.); hajeena.saravanapavan@nhs.net (H.S.); john.barragry@nhs.net (J.B.); jennifer.rogers6@nhs.net (J.R.); shameer.mehta@nhs.net (S.J.M.); 2Department of Dietetics, University College London Hospitals NHS Foundation Trust, 250 Euston Road, London NW1 2PG, UK; k.keetarut@nhs.net; 3UCL Medical School, 74 Huntley Street, Bloomsbury, London WC1E 6DE, UK; hui.koo.14@ucl.ac.uk

**Keywords:** enteral nutrition, Crohn’s disease, exclusive enteral nutrition, partial enteral nutrition

## Abstract

Medical and surgical treatments for Crohn’s disease are associated with toxic effects. Medical therapy aims for mucosal healing and is achievable with biologics, immunosuppressive therapy, and specialised enteral nutrition, but not with corticosteroids. Sustained remission remains a therapeutic challenge. Enteral nutrition, containing macro- and micro-nutrients, is nutritionally complete, and is provided in powder or liquid form. Enteral nutrition is a low-risk and minimally invasive therapy. It is well-established and recommended as first line induction therapy in paediatric Crohn’s disease with remission rates of up to 80%. Other than in Japan, enteral nutrition is not routinely used in the adult population among Western countries, mainly due to unpalatable formulations which lead to poor compliance. This study aims to offer a comprehensive review of available enteral nutrition formulations and the literature supporting the use and mechanisms of action of enteral nutrition in adult Crohn’s disease patients, in order to support clinicians in real world decision-making when offering/accepting treatment. The mechanisms of actions of enteral feed, including their impact on the gut microbiome, were explored. Barriers to the use of enteral nutrition, such as compliance and the route of administration, were considered. All available enteral preparations have been comprehensively described as a practical guide for clinical use. Likewise, guidelines are reported and discussed.

## 1. Background

About 115,000 people in the UK have Crohn’s disease (CD) [1,2]. Up to a third of patients are diagnosed during adolescence, leading to many years of disease and associated morbidity. Half of these patients will require surgery during the natural history of their disease, and more than 75% of those will undergo further surgery [3,4,5]. Complications such as stricturing or penetrating disease with fistulae and abscess formation occur in a significant proportion of patients. Genetic and environmental factors affect CD phenotypes [6,7]. CD phenotypes in childhood are more aggressive than when the disease arises in adulthood [6,7]. 

In the last two decades many drugs have been developed to treat CD and a ‘top-down’ approach is now favoured to achieve sustained clinical remission and resection-free survival for patients with CD [8]. Many patients do not respond to currently available treatments (immunosuppressants, biologics, and corticosteroids) or have an initial response that is not sustained. Efficacy also differs if a therapeutic agent is administered in naïve patients with newly diagnosed CD, or after failure/loss of response to other treatments [9,10]. 

Treatments are associated with important toxic effects, such as infections or even malignancy risks. The goals of CD therapy include mucosal healing which affects the natural history of the disease but also avoidance of complications and improvement of quality of life and are achievable with biologics, immunosuppressive therapy and specialised enteral nutrition, but not with corticosteroids [11,12]. Sustained remission remains a therapeutic challenge. Cost of treatment, particularly in the context of a ‘top down’ approach, may prohibit access to therapy for patients in many countries where there is no central funding [13]. 

### 1.1. Exclusive and Partial Enteral Nutrition

Enteral nutrition (EN) containing macro- and micro-nutrients are nutritionally complete high energy artificial nutritional supplements in powder or liquid form [14]. Exclusive enteral nutrition (EEN) provides 100% of daily nutritional requirements from a liquid nutrition formula either orally or via a feeding tube. When 35–50% of habitual food intake is replaced with artificial enteral nutrition, such as oral nutritional supplements (ONS), it is known as partial enteral nutrition (PEN) [15]. 

EN is a low-risk, non-invasive therapy. It is well-established and recommended as a first line induction therapy in paediatric CD with remission rates of up to 80% [16]. Other than in Japan (and increasingly in China), EN is not routinely used in the adult population of Western countries, mainly due to unpalatable formulations that lead to poor adherence [17]. 

In the 1970s, Voitk et al. [18] described the value of EN in improving nutritional status (support value) and inflammatory markers (therapeutic value) in active inflammatory bowel disease (IBD). Despite the advances made in CD algorithm therapy and increased interest in nutrition-based treatments, the role of EN is still undetermined and it is rarely recommended by gastroenterologists as first-line therapy. As such, EN is under-utilized. Additionally, the optimal formulation or diet to follow during PEN is still to be determined. Variation exists in the uptake of EN worldwide with Europe, Australasia and Canada showing higher use particularly in the paediatric population, while the uptake remains low in adults [19]. 

EN has been used prior to elective surgery in order to reduce surgical complications. It has been reported to down-stage the need for urgent surgery and reduces post-operative complications such as an anastomotic leak or abscesses, by reducing inflammation, improving nutritional status, and decreasing the antigenic load through bowel rest [20]. 

EN has been shown to influence gut microbial diversity, an important component of CD pathogenesis which may restore gut homeostasis and prolong remission, as well as improve our understanding about the aetiology and trigger factors of CD [21,22]. However, dropouts in EN trials are frequent due to unpalatable formulations and poor acceptance of a nasogastric tube [23]. 

### 1.2. Objectives and Methods

This study aims to offer a comprehensive review of available EN formulations and the literature supporting the use and mechanisms of action of EN in adult CD patients, to support clinicians and patients in real world decision-making when offering/accepting treatment. A literature search (Medline, Embase, Ovid, and Cochrane Database) was carried out from inception to May 2019 in order to identify both randomised and non-randomised studies that reported the efficacy of EEN or PEN for the induction and/or maintenance of remission in CD, used either as a single therapy or in combination regimens, in adult Crohn’s patients. The mechanisms of action of enteral feed including their impact on gut microbiome are explored. Barriers to the use of EN such as adherence and the route of administration are considered. All available enteral preparations are comprehensively described as a practical guide for clinical use. Likewise, guidelines are reported and discussed. 

## 2. EN Formulations

### 2.1. Composition of EN Formulae

The EN formulations used in the UK vary depending on whether they are used orally or via an enteral tube (Table 1). EN can be nutritionally complete or incomplete, reflecting the reference values for macro and micronutrients for a healthy population. Nutritionally complete EN has a balanced nutritional composition of macro-and micronutrients, including essential fatty acids and micronutrients and essential amino acids, and therefore can be used as the sole source of nutrition for prolonged periods. Oral nutritional supplements are usually in liquid form but they are also available in powder and dessert-style. Feeds via a tube are usually nutritionally complete in a 1.5 L bottle [24,25]. Standard EN usually contains whole protein formulas (intact proteins), lipids (mostly long chain triglycerides), and carbohydrates (predominantly as polysaccharides, e.g., maltodextrin) [25]. They can be fibre or non-fibre containing formulae and are usually free of lactose and gluten [24,25]. 

### 2.2. Composition Types

The composition of EN is classified by the nitrogen source derived from the amino acid or protein component of the formula. There are three main types of EN currently available on the UK market, namely elemental (amino-acid based), semi-elemental (oligopeptide) and polymeric (whole protein) feeds [23] (Table 1). 

Elemental feeds are used in cases of severe malabsorption, or where there may be impairment of the gastrointestinal tract, or in cases of milk protein allergy. They are created by mixing free single amino acids to meet protein requirements. They are monomeric, low molecular weight, chemically defined formulae and are entirely antigen-free [23,24]. The main feeds are E028 extra liquid, E028 extra powder and Emsogen powder. Emsogen has the additional advantage of a higher medium chain triglyceride (MCT) content. 

Oligopeptide or semi-elemental diets are peptide-based formulae made by protein hydrolysis with protein predominantly in peptide form. The mean peptide chain length is four or five amino acids, which is too short for antigen recognition or presentation [23,24]. In severe cases of CD where malabsorption can occur, formulas containing peptides and medium chain triglycerides can facilitate absorption and may be recommended over polymeric feeds [25]. Common UK-based feeds include Peptamen^®^ and Vital 1.5^®^. 

Finally, polymeric feeds contain whole proteins from sources such as milk, meat, egg, or soy. The composition of these feeds makes them more palatable and they are therefore commonly used to optimise compliance [26]. The most commonly used feeds used for CD include Modulen IBD ^®^, Ensure plus Milkshake style^®^, and Fortisip^®^. 

### 2.3. Evidence

There are proposed advantages of Modulen IBD^®^ which has been specifically designed for use in CD patients in order to induce disease remission. It is 100% casein-based and contains a naturally occurring transforming growth factor β2 (TGF β2), an anti-inflammatory cytokine which may help in the inflammatory response and may assist with gut mucosal healing. TGF β2 has been demonstrated to be an important mediator of mucosal defence and repair in mice [27], and in humans treated with EEN [28,29]. The direct therapeutic impact of TGF β2 within Modulen IBD^®^ remains to be proven. Multiple studies have demonstrated successful results using EEN for the induction of remission using Modulen IBD^®^, but these have predominantly been conducted in paediatric CD [28,30,31,32,33] with no studies specifically stating the use of Modulen IBD^®,^ for the induction of remission in adult CD. 

Despite there being a wealth of literature supporting the use of EEN in CD, the exact mechanism of action and ideal formulation remains unknown [23]. The decision on the type of formula used may be influenced by evidence from clinical studies, the dietary modulation of the intestinal immune response in IBD, and its potential clinical implications [34]. There is currently insufficient evidence to make firm recommendations about one feed composition over another [35,36]. Therefore, standard polymeric feeds are often utilised due to increased palatability [37,38,39,40]. 

A recent Cochrane review demonstrated no difference in remission rates or side effects between elemental, semi-elemental and polymeric EN formulas [23]. Three of these studies compared EN formulations with different fat composition but similar protein composition, and one study compared non-elemental formulas differing in glutamine enrichment [23]. Subgroup analyses indicated there was no statistically significant difference in remission rates between trials comparing elemental and polymeric feeds [37,38,39,40,41,42,43] or between trials comparing elemental and semi-elemental diets [44,45,46,47]. 

Verma et al. [48] found no difference between elemental and polymeric feeds and advantage to facilitate steroid withdrawal or reduce need for surgery. However, tolerance to elemental formula could be a limitation. In this study, 32% (6/19) of participants in the elemental EN group withdrew from the study due to poor tolerance of the formula because of taste or smell. 

Despite the lack of a recommended feed, composition studies have demonstrated a non-significant trend favouring low fat formulations [45,49] with the trend supporting LCT content which was strongest for the lowest value evaluated (<5% LCT) [23]. Therefore, a recommendation may be made to consider the use of EN with a lower fat content [34]. Several studies have hypothesized that the proportion or type of fat in EN can affect the production of pro- or anti-inflammatory mediators [47,50,51,52]. Very low-fat content (<3 g/1000 kcal) and very low LCT formulations induce higher remission rates than higher fat content EN formulas [23]. Ajabnoor and Forbes [53] conducted a systematic review and found that feeds low in LCT, high in MCT, low in monounsaturated fatty acids, and those rich in linoleic acid, such as safflower oil, are all associated with higher response rates to EN. Interestingly, the total n-6 fatty acid content positively correlated with response rates. The correlation was significant when expressed as the ratio between n-6 and n-3 fatty acids. A non-significant positive trend was founded between MCT delivery as a percentage of the total energy provision and response rate. Conversely, there was a non-significant negative trend for the content in monounsaturated fatty acids. However, there was not any statistically significant difference in remission rates between low fat (< 20 g fat/1000 kcal) and high fat (>20 g fat/1000 kcal) feeds [23]. 

## 3. Oral Versus Tube Feeding

EN comprises all forms of nutritional support delivered into the gastrointestinal tract, independent of the route of application, and can be provided orally or via an enteral tube [24]. Nasogastric and nasojejunal tubes are inserted via the nose; enteral tubes can also be inserted via a stoma inserted endoscopically into the stomach, i.e., percutaneous endoscopic gastrostomy or with a jejunal extension, or inserted into the jejunum (percutaneous endoscopic jejunostomy). Each of the aforementioned enteral tubes may also be placed surgically, i.e., via surgical gastrostomy or jejunostomy [25,54]. 

Decisions about the optimal route for the delivery of artificial nutrition in IBD require consideration of several aspects, including the ability of the patient to eat, the anatomy (especially post-surgery), absorptive capacity of the gastrointestinal tract, the nutritional status of the patient, and the therapeutic goals (whether these are for treatment of malnutrition, induction or maintenance of remission or for nutrition support only) [34]. 

The oral administration of polymeric formulations, where compliance may be higher, has been demonstrated to be as efficacious as feeds delivered by the nasogastric route [55]. The ability to use a palatable oral polymeric formulation as a means of induction therapy in CD helps to overcome some of the prior limitations of EN treatment [55]. If the oral administration of EN feeds is inadequate to meet nutritional requirements, enteral tube feeding can be used to optimize compliance and effectiveness [26]. Therefore, it is hypothesised that, when used for the induction of remission, it is the adequate administration of the feed rather than the route of delivery that determines the efficacy. 

There are cultural variations in both the use of EEN and route of administration. In adult CD patients EEN is not regularly used as the first line therapy especially in Western countries, while it is a standard practice in Japan and increasingly in China [56,57]. These differences may be the result of differences in financial schemes and professional medical attitudes. For example, in Japan where fees for EEN therapy are covered by the national insurance scheme, EEN is recommended as the primary therapy, and as a result, patients take elemental nutrition by a nasogastric tube [56]. Conversely, cost is a main barrier for patients in the USA and Canada [58]. The European Society of Clinical Nutrition and Metabolism (ESPEN) 2017 guidelines [34] suggest using oral feeding as the first step for EEN treatment and turn to nasogastric tube feeding if oral intake is insufficient, taking feasibility and effect on quality of life into account [59]. Research on the global variations of EEN use in children with CD shows that about 66% of patients prefer oral to nasogastric administration [58]. Reasons for the lack of uptake among physicians include concerns about the poor acceptability of nasogastric tubes by patients, palatability of formulations, and lack of compliance with a restrictive dietary intervention [60]. The large volume needed, long duration, together with physical and psychological influences resulting from nasogastric tube insertion, and the physical presence of the nasogastric tube, may explain why more patients are unwilling to use this route [61]. 

PEN is the replacement of 35–50% of habitual food intake with EN. The provision of PEN is routinely used in research studies in Japan to prolong disease remission by delivering a portion of caloric intake via overnight enteral tube feeding and allowing patients to eat normally during the day [62,63,64]. 

Joly et al. [65] tested the absorption among adult patients with short bowel syndrome after surgery by oral or nasogastric feeding alone, and the two routes combined demonstrating that oral feeding alone had less net absorption of macronutrients than the other two methods. Rubio et al. [55] compared fractionated oral with continuous enteral feeding over eight weeks for paediatric CD patients and found no significant difference in induction of remission. The nasogastric fed group had greater weight gain than the oral feeding group. However, this study was a retrospective trial without proper randomization, thus limiting its generalisability. This aspect also warrants the need for more studies specifically addressing the delivery mode of enteral nutrition. 

In healthy adults, enteral tube feeding as continuous and bolus feeding has been shown to have little effect on the appetite and leads to a significant increase in total energy intake [66,67]. This is presumably due to bypassing the cephalic phase of digestion in the upper gastrointestinal tract and suggests that enteral tube feeding could be used as a successful way to increase weight by significantly increasing total energy intake versus an oral diet alone. Responses in terms of appetite to a single meal cannot necessarily be extrapolated to long term gains, while the issues of compliance and weight gain maintenance remain, with reports suggesting that a multidisciplinary team approach may overcome this obstacle [19,68]. 

## 4. Mechanisms of Action

While the mechanisms of action of EN are still not fully understood, it is thought to interact with key aspects of the pathophysiology of CD through (a) modulation of the pro-inflammatory state and (b) alteration of the gut microbiome [17]. 

### 4.1. Modulation of Pro-Inflammatory State

EN has been found to suppress various factors involved in the heightened inflammatory pathways in CD. Clinical trials involving EEN in patients with active disease have found significant drops in mucosal inflammation both macroscopically and histologically. These were associated with reduction in levels of pro-inflammatory cytokines such as IL-1β, IL-1ra, IL-6, IL-8 and TNF-α [28,69]. Analysis of microRNA expression after EEN also showed a reduction in those associated with inflammation, with a closer resemblance to expression profiles from healthy controls [70]. 

Further in vitro studies have found that culturing samples with EN resulted in down regulation of IL-6 and IL-8 [71,72]. This is thought to occur through modulation of gene expression in the NF-κB and p38 mitogen protein kinase pathways [71,73]. Dietary components, such as glutamine and arginine, likely play a role through blocking phosphorylation in signalling cascades, while other elements including vitamin D3 and α-linoleic acid are also believed to contribute [74]. 

### 4.2. Alteration of Gut Microbiome

Another important feature of CD is dysbiosis, an imbalance in the composition of gut microbiota. It is presently unclear whether this imbalance is a contributing factor or consequence of CD [75,76]. There appears to be a depletion of the phyla Firmicutes, containing anti-inflammatory symbionts, and an expansion of the phyla Proteobacteria, containing pro-inflammatory pathobionts [77,78], in addition to a probable loss of bacterial diversity [79]. 

Significant changes in microbiota composition have been shown to occur in CD after treatment with EN, in contrast to healthy subjects where they remained relatively constant [32,80]. The taxonomic shifts noted across various studies found a fall in Firmicutes, Bacteroides/Prevotella and Proteobacteria, with an increase of Bacteroidetes [81]. Paradoxically, this shows that there is probably a trend towards microbial imbalance with a further reduction in taxonomic diversity after treatment with EN [82,83,84]. It has been proposed that EN exerts its effect by depleting harmful bacteria allowing for subsequent re-colonisation [81,85]. In addition, functional diversity which is elevated in CD, is reduced to normal levels with the use of EN [83,86]. 

EN also appears to influence bacterial metabolism which is altered in CD. Studies of faecal samples have shown reduction of metabolites such as short-chain fatty acids, 1-propanol, 1-butanol, esters of short-chain fatty acids and butyrate [82,87]. Assessment of community level function through metagenomic sequencing found that EN resulted in upregulation of genes in the spermidine/putrescine biosynthesis pathway involved in cell growth, and downregulation of genes involved in biotin and thiamine synthesis [83]. 

An additional impact of EN could be due to the interaction of the two mechanisms. Tight junction integrity of the intestinal epithelium is modulated by the protein expression of myosin II regulatory light-chain kinase through the NF-kB pathway [88,89,90]. Downregulation of this pro-inflammatory pathway reduces intestinal permeability and prevents the adherence of invasive bacteria and trafficking of immunogenic substances. This suppresses the vicious cycle of inflammation and bacterial penetration [85,91]. 

## 5. Guidelines

In this section we describe the guidelines currently in use for EN in CD. Table 2 outlines the main points. We next describe ESPEN 2017 guidelines [34]. The recommendations for use on EN in CD in the ESPEN guidelines [34] can be divided into three sections: use in active disease, maintaining remission, and peri-operatively. 

In active disease, there are four main recommendations. The first is that if artificial nutrition is indicated, EN is always preferable to parenteral nutrition (PN). Oral feeding and ONSs are the first steps, and enteral tube feeding used if oral feeding is not possible or insufficient. They suggest EN should be considered in those with a functional gastrointestinal tract who are unable to swallow safely. Even if there is an impaired absorptive ability, EN should be tried along with PN supplementation [92,93,94]. The second relates to the induction of remission. EN is recommended as first line treatment in the paediatric population while no clear recommendation was made for adults. This is on the basis that the importance of growth in children and adolescents makes preventing undernutrition and avoiding the use of steroids a considerable priority [26,95,96,97,98]. The third is that specific formulations are not recommended and standard EN (polymeric, moderate fat, no particular supplements) can be used for primary or supportive nutritional therapy. This is due to insufficient evidence of statistically significant difference between types of enteral formulae in terms of protein composition (elemental, semi-elemental, oligomeric, polymeric), fat content or addition of growth factors [35,36,62,99,100,101,102]. Lastly, for administration of EN, enteral tube feeding can be by nasal or gastrostomy tubes and continuous enteral feeding via a pump should be used instead of bolus feeding, given the lower complication rates seen in continuous feeding [103,104,105,106,107]. 

For maintaining remission, there are two main recommendations. The first is that EN is not recommended as a primary therapy to maintain remission, but ONS/EN can be used for malnutrition. Due to the heterogeneity of published studies, with most pre-dating new maintenance treatment modalities, it was difficult to recommend using EN as a first line treatment [34]. The second is that a standard formulation should be chosen if EN is used. The guideline found insufficient evidence on the difference between elemental and polymeric EN formulations to make recommendations. However, the adherence rate for elemental formulas was shown to be lower than an unrestricted diet or polymeric formulas, possibly due to lower palatability and higher cost [43,108]. 

In the peri-operative use of EN, there are three main recommendations. The first is that if a patient’s food intake is insufficient to meet their energy needs, ONSs should be encouraged. If this is insufficient, they should be given EN (or PN if EN is contraindicated). The second is that pre-operatively, if malnutrition is diagnosed, surgery should be delayed 7–14 days where possible and the patient should be intensively artificially fed. For emergency surgery, artificial nutrition should be initiated at the time if the patient is malnourished, or if an oral diet cannot be recommenced within seven days post-surgery. The third is that post-operatively, early nutritional support regardless of route should be commenced within 24 h post-surgery. These recommendations are all based on evidence from studies on nutritional status and operative outcomes. Nutritional support to prevent and treat undernutrition, both before and after surgery, has a well-documented effect on post-operative complications, morbidity, and mortality [109,110,111,112,113,114,115,116,117,118]. 

## 6. Efficacy of EEN and PEN in Induction and Maintenance of Remission (Monotherapy and Combination Therapy)

### 6.1. EEN in Treatment of CD

EEN has been shown to be associated with a higher rate of mucosal healing, altered intestinal flora, enhanced bone turnover, better quality of life and improved nutritional status (see Table 3 for studies). A recent small non-randomised prospective feasibility pilot study conducted in New Zealand showed that two weeks of EEN in 38 patients aged 16–40 years, significantly improved clinical symptoms, serum c-reactive protein, insulin-like growth factor 1 (marker of nutritional status) and faecal calprotectin [124]. During the ‘induction’ period patients could only drink multiple cartons of 200 mL Ensure plus (Abbott laboratories, the Netherlands), which is a more palatable polymeric formula and additional fluids daily. 

Treatment progressed with six weeks of either EEN or PEN. Both groups sustained the initial improvements in nutrition, disease symptoms and markers of inflammation, suggesting that PEN (which included one balanced social meal a day) is equivalent to EEN following the ‘induction’ period, potentially improving compliance and consequently uptake of therapy if the use of EEN is for a limited period of two weeks-time. After the eight weeks of treatment both groups reintroduced usual food and drinks over a period of three days. Contact with the dietitian was scheduled at weeks 0, 2, 4, 6, and 8, suggesting that patients require continuous support to adhere to EN regimens. Interestingly, nutrient and energy intake during PEN were calculated using an app, showing that new technologies may provide the necessary support. BMI decreased in both EEN and PEN groups but with only minimal changes in the latter. Unexpectedly, compliance was higher in patients over 18 years of age compared to adolescents. Specifically, 73% of adolescents versus 26% of young adults dropped out of EN treatment. The authors concluded that EN is effective and might be of interest to a subgroup of patients, such as motivated young adults who may choose nutritional therapies. Nevertheless, the lack of a non-EN control group limits the relevance of these findings [124]. 

Early improvement of disease activity has been observed in other studies in the adult population. Historically, EN has been compared to corticosteroids for rapid efficacy during the acute phase showing that corticosteroids were more effective than EN in treatment of active CD [68]. In particular, corticosteroids are frequently used in primary and secondary care to treat flares of CD and as a bridge to control symptoms until other prescribed therapies become effective (e.g., immunosuppressants). Corticosteroid resistance and dependence occur in 8–22% and 15–36% of patients, respectively [68]. 

A Cochrane meta-analysis published in 2007 analysed data from six studies of which one was in a paediatric cohort and favoured the use of corticosteroids in induction of remission [125]. Nevertheless, when the analysis was restricted to only high-quality studies, there was no difference between corticosteroids and EEN. A potential explanation for their contrary findings may be that they combined data from both the adult and paediatric populations [125]. 

In routine clinical practice, corticosteroids are preferred to EN as induction therapy and it is unclear if EN is offered as an option to adult patients or who takes this decision. 

An update of the Cochrane meta-analysis, published in 2018 by the same scientific group, confirmed the previous findings [23]. Unlike the previous systematic review and meta-analysis, the authors not only performed an intention-to-treat analysis but also included a per-protocol analysis excluding lack of efficacy due to withdrawal caused by poor palatability and non-acceptance of a nasogastric tube. 

The aim of this meta-analysis was to evaluate effectiveness and safety of EN as primary therapy for induction of CD and included studies up to 5 July 2017. They compared EN formulations among them and EN vs. corticosteroids. Results were pooled in subgroups based on formula composition and age. 

The meta-analysis was conducted on 11 papers (378 subjects both adults and children) and showed no differences in induction of remission. Even when data were stratified based on enteral diet composition (elemental, semi-elemental and polymeric) the results were confirmed. No difference was found in remission rates of patients in the elemental group (64%) compared to 62% in patients in the non-elemental group. Both adverse effects and withdrawals due to adverse events were un-related to which EN feed the subjects were receiving. The most common side effects were nausea, vomiting, diarrhoea and bloating [23]. 

When analysing the efficacy of EN in adults compared to corticosteroids, a difference was noted with corticosteroids being superior to EN (45% (87/194) in adult patients compared to 73% (116/158) of corticosteroid patients; risk ratio (RR) 0.65, 95% CI 0.52–0.82)). Interestingly, there was no statistically significant difference in remission rates between EN and corticosteroids therapy on a per-protocol analysis suggesting that compliance is key in these findings. Adverse events were incurred in both groups with no difference. However, patients receiving EN were more likely to withdraw due to side effects than those on corticosteroids therapy. In the EN group the most common side effects were heartburn, flatulence, diarrhoea and vomiting, and moon facies, dysglycaemia and muscle weakness in the corticosteroid group [23]. 

This meta-analysis shows that the inability to tolerate EN persisted as the most common reason for withdrawal, again suggesting that the place for EN in CD is still to be defined in a subgroup of selected motivated patients. In fact, the results were all classified as very low-quality data including the superiority of corticosteroids compared to EN in induction of remission of CD in adults. The authors concluded that industry should focus in providing a choice for more palatable formulations to be delivered without the use of a nasogastric tube and increase adherence to EN. However, the latest meta-analysis was not powered to define the optimal route of administration of EN. 

At sub-analysis protein content or composition did not affect results while a very low-fat content (<3 g/1000 kcal) and very low long chain triglycerides were associated with higher remission rates than other EN formulations, which may guide choice of formulation. Nevertheless, the authors rated the results as very low-grade quality evidence. 

Of note, the effect of fat composition in EN has been evaluated in a systematic review by Ajabnoor and Forbes [53] that included 29 trials, concluding that there was a positive association between the total n-6 fatty acid content and response rate particularly for the ratio of n–6 and n–3 fatty acids (r = 0.378, *p* = 0.018). 

Individual studies have shown a wide range of efficacy of EN ranging from 20% to over 80%. For example, in Japan, EN has been accepted as the first-line therapy in patients with active and quiescent CD achieving efficacy of over 80%. Perhaps this is the result of the flexibility in administering EN in Japan, which is known as the “slide method” of facilitating compliance [56,93,123,126]. Flexibility in accessing therapy is key since patients increase the amount of enteral feed when the disease activity worsens or decrease it when responding to therapy. If bowel obstruction or septic complications are not present, active disease is treated by EEN (nasogastric tube feeding). Once induction of remission is achieved low fat foods are allowed in addition to EN, and the proportion of calories derived from foods is gradually increased. In clinical remission, approximately half of the calories are provided by EN at their home. 

The latest studies and reports have demonstrated that disease phenotype does not affect efficacy of EN (i.e., small bowel vs. colonic disease) and therefore for example in children EN is indicated for any type of active luminal CD [43,120,127,128,129,130]. 

The effect of EEN on health-related quality of life in adult individuals with active CD has been examined by Guo et al. [57]. Patients (n = 13) received a polymeric feed orally during the day and via a self-intubated nasogastric tube at night for four weeks. The inflammatory bowel disease questionnaire was used to evaluate the impact of EEN on health-related quality of life and showed a significant improvement in the score in all dimensions such as bowel symptoms, social function, systemic symptoms and emotional status. Remission was achieved in 11 patients (84.6%). Patients were also asked two more specific questions regarding EN such as if they would have wanted to receive EN for a longer time and being treated with EEN if the disease relapsed again. Patients responded in 61.5% of cases that they would like to receive EEN again if the disease relapsed—only 15.4% expressed a willingness to receive EEN for a longer period [57]. 

### 6.2. PEN in Induction of CD Remission

PEN has been studied in the induction of remission by Sigall-Boneh et al. [133]. In their study, PEN (up to 50% of calories from a polymeric formula) with a specific diet, excluding components hypothesized to affect the microbiome or intestinal permeability, effectively induced disease remission in 70% of children and 69% of young adults of the total 47 subjects with early mild to moderate luminal CD included. Improvement of inflammatory markers was noted in 70% of patients in remission. When the CD-specific exclusion diet was used alone rather than in combination with PEN, remission was achieved in six out of seven patients. Despite this finding, PEN has not been established as an induction treatment. Besides, the study did not include an EEN control group and the remission achieved with the exclusion diet alone suggests that it may have a role alone rather than in combination with PEN [133]. 

### 6.3. PEN in Maintenance of CD Remission

A Cochrane review on the use of EN for maintenance of remission in CD included both EEN and PEN and was published in 2018 [35] as an update of the previous review published in 2007 [134]. The aim of the review was also to assess the impact of different formulations on outcomes and their safety profiles. In total, four randomised controlled trials including 262 adult individuals were selected. Elemental diet was compared to polymeric diet in one study (no difference in remission at 12 months), to normal diet (PEN group had a lower chance of relapsing at 12 months) in another one, and to 6-mercaptopurine (no difference in remission rates at 12 months between groups) or no treatment in a third study [135]. The subjects that received immunosuppression experienced hair loss, liver injury and an abscess. In the group receiving the elemental diet, no adverse events were observed except surgery due to worsening of CD. In the fourth study a polymeric diet with a specific formulation rich in TGF β2 was compared to mesalamine finding no difference in remission rate at 6 months and with reported side effect in the EN group of nausea and diarrhoea [136]. Side effects in the mesalamine group were not reported. Data were not pooled due to heterogeneity in interventions and assessment of outcomes. Interestingly, the authors concluded that the outcomes evaluated in the review were uncertain and no conclusions could be drawn on the safety and efficacy of EN in maintenance of CD. 

Takagi et al. [62] compared PEN (half elemental diet) used in 26 patients to an unrestricted diet in 25 patients as a maintenance strategy for individuals who underwent induction of remission with EN, corticosteroids and/or infliximab. Use of azathioprine was allowed if already established on this treatment. In addition, patients were instructed to take oral mesalamine. After a mean follow up of 11.9 months, 64.0% of subjects receiving PEN were in remission compared to 34.6% of individuals on a free diet (multivariate hazard ratio 0.4, 95% CI: 0.16–0.98). 

Another study evaluating the efficacy of supplemental polymeric to elemental formula (providing between 35% and 50% of calorie intake) in addition to unrestricted normal diet in 33 steroid-dependant subjects, for maintenance of CD, did not find any significant differences including in facilitating withdrawal of corticosteroids therapy [137]. Immunosuppressant and mesalamine were permitted and continued if patients were already established on these regimens. 

These two randomized controlled trials were analysed in the Cochrane review by Akobeng et al. [35], but data analysis was limited by small sample sizes. A systematic review of 10 studies concluded that PEN with polymeric or elemental formulation significantly increased length of remission compared to no PEN [138]. However, the majority of studies where EN has been shown to reduce incidence of disease recurrence [11,62], post-operative recurrence [101,102], improve drug efficacy [139,140,141,142] and be as effective as 6-mercaptopurine in maintenance of remission in adult CD for up to two years [135] have been conducted using elemental formulation administered via a nasogastric tube. EN was not found to impair quality of life in CD [143]. As these studies were all conducted in Japan and China it is difficult to generalise these findings at present, until further larger studies are conducted in different population groups. 

PEN is currently recommended in the clinical practice guidelines in Japan for at least 1 year in patients in whom remission was induced by EN and elemental formulation is recommended as effective in maintaining remission in CD [144]. Due to limited evidence, PEN is not currently recommended in any UK or European clinical guidelines for the maintenance of remission for CD [34,54,59,107]. This is despite offering the potential of reducing the need for immunosuppressive therapy and related side effects. 

Recent studies show that PEN is effective in maintenance of remission in CD. Heterogeneity and quality of data and small sample sizes are limitations. Compared to a habitual diet, PEN has been shown to lengthen disease remission and reduce post-operative recurrence without impairing quality of life [15,108,145]. 

### 6.4. Combination Therapy

With the advent of biologics, the role of EN has been reassessed in combination with Infliximab in moderate to severe CD. Combination therapy with infliximab and PEN was more effective than infliximab monotherapy both in inducing and maintaining remission [140]. There are no data on use of PEN in combination with the other currently available biologic drugs. The current strategy when patients lose response to infliximab maintenance therapy is either to shorten the interval of infliximab administration, increase the dose or replace infliximab with another biologic agent. Another option is to use concomitant immunomodulators. The safety profile and the adverse events which may occur with these medications need to be taken into consideration when offering treatment to patients. 

In a multicentre, retrospective, cohort study by Hirai et al. [141], the effectiveness of EN to enhance infliximab-induced maintenance of remission was studied. One hundred and two adult CD patients from seven centres in Japan receiving infliximab maintenance therapy every 8 weeks were included. In total 45 patients were in the EN group and 57 in the non-EN group. The EN group had a significant advantage in terms of cumulative remission rate compared to the non-EN group (*p* = 0.009). Recurrence occurred in 14 (31.1%) of the 45 patients in the EN group during the follow-up period. In comparison, 33 of the 57 patients (57.8%) in the non-EN group had recurrence. Risk factors for recurrence were evaluated using a Cox proportional hazards model, considering such various characteristics as concomitant drugs, disease location, and behaviour. EN (900 kcal/day) was the only factor to prevent disease recurrence (*p* = 0.01) according to multivariate analysis [141]. No PEN-related adverse events were reported. These findings suggest that combination therapy with infliximab and EN improves the maintenance of remission with infliximab. In addition, EN has non-pharmaceutical properties which translate into a favourable safety profile. Of note stenotic disease was more common in the EN group, while there were more smokers in the non-EN group [141]. 

Conversely, Yamamoto et al. [100] found no benefit of EN combined with infliximab as maintenance therapy in a single-centre prospective study. In the EN group, 32 of the 56 patients who achieved clinical remission with infliximab induction therapy received infliximab as maintenance therapy and concomitant EN—with an infusion at night of the elemental diet (1200–1500 kcal/day) and a low-fat diet during the daytime. The non-EN group (24 patients) received neither nutritional therapy nor food restriction. Lipid content have been shown in several trials to affect efficacy due to their characteristic of acting as precursors of inflammatory mediators. Both small bowel and colonic disease was present in 34 individuals. During the 56-week follow up period, in this subgroup, 18 of 21 patients (86%) in the EN group and nine of 13 (69%) in the non-EN group maintained clinical remission (*p* = 0.47). The sample size of their study was small with a short follow-up which may account for the negative findings. The authors concluded that concomitant EN and infliximab maintenance therapy in quiescent CD did not statistically add to the maintenance rate of clinical remission. In the EN group both the mean CDAI score and the cumulative remission rate were more favourable. It may be assumed that a larger sample size would have produced a significant difference between the EN and the non-EN group. Therefore, in order to determine the effect of EN in combination with infliximab maintenance therapy, a larger multicentre, randomized, controlled trial is required. 

The authors of both studies did not provide a complete explanation of the discrepancy between the results for the efficacy of concomitant EN therapy during infliximab maintenance therapy. It can be speculated that the timing of when infliximab is used affects the efficacy. Specifically, the duration of the disease is critical. In the study by Yamamoto et al. [100], mean disease duration before entry into the EN and non-EN groups was 33 ± 4.4 and 35 ± 4.0 months, respectively. Disease-duration data are lacking for the subjects in the study by Hirai et al. [141]. 

Concomitant use of EN and infliximab in the maintenance of remission has also been studied by Sazuka et al. [139] in 74 patients who had successful infliximab induction therapy. The use of 600 kcal/day of elemental and/or polymeric formulas was identified as an independent factor associated with sustained response to infliximab. The authors concluded that concomitant EN treatment with 600 kcal/day reduced the incidence rate of loss of response to infliximab (20.6% versus 52.3%). 

Those findings were confirmed in a recent meta-analysis that included four studies (n = 342) and evaluated the benefit of infliximab and EN combination therapy with infliximab monotherapy for the induction and maintenance of clinical remission [140]. Patients were either treated with infliximab and EN (n = 157) or infliximab monotherapy (n = 185). Patients were supplemented with at least 600 kcal/day with elemental or polymeric nutritional feeds. Infliximab monotherapy was significantly inferior to infliximab and EN combination therapy both in inducing [69.4% versus 45.4%; odds ratio (OR) 2.73; 95% CI: 1.73–4.31, *p* < 0.01] and maintaining remission (74.5% versus 49.2%; OR 2.93; 95% CI: 1.66–5.17, *p* < 0.01) at one year. Based on their results, four patients were needed to treat with EN to maintain clinical remission. There was no heterogeneity between studies [140]. 

### 6.5. EN in Complicated and Extra-Intestinal CD

Evidence in the literature suggests that EN has a role beyond the treatment of luminal disease. EEN is also useful in the setting of complicated CD and extra-intestinal CD as suggested by the increasing number of publications in recent years. However, this evidence comes from small reports or retrospective analyses. Further evidence is required to establish the use of nutritional therapy in complicated CD. CD can affect any segment of the gut, from the oral cavity to the anus and can cause transmural inflammation. From the initial luminal inflammation, CD can progress to strictures or penetration complicated by fistulae and abscesses. 

Many of these conditions could lead to intestinal insufficiency or intestinal failure, potentially necessitating parenteral nutrition as the chosen feeding route [98,146,147,148]. If clinically permitted, EN should be preferred as the first feeding route, since its use has a trophic effect on the gut and stimulates bowel motility [34]. In terms of adverse effects, parenteral nutrition can be associated with higher morbidity compared to EN, in terms of sepsis, catheter-related blood stream infections, thromboses, and liver dysfunction [34,145,149,150]. 

#### 6.5.1. Penetrating CD

Two recent Chinese studies demonstrated substantial benefits of EEN in the management of fistulizing CD [151,152]. Yang et al. [151] included 33 patients with entero-cutaneous (4 patients) or entero-enteric fistula complicated by abscess formation. Disease activity was evaluated by endoscopy, biochemistry, and disease activity scores before and after EEN. Clinical remission was achieved in 27 out of the 31 patients (80.5%) who completed 12 weeks of EEN. In total, only three patients required surgery. About half of the cohort (43%) had perianal disease when starting EEN however the effectiveness in perianal disease was not reported. Fistula closure occurred in three out of four (75%) of the patients. To give some perspective on these results, the fistula closure rate with infliximab is 68% [153]. In patients with entero-enteric fistula with associated abscess, a resolution of the abscess was noted in 76% of cases. Antibiotics were concomitantly used but not drainage in individuals with abscesses. 

In the study by Yan et al. [152], 48 patients with entero-cutaneous fistula were treated exclusively with 12 weeks of EEN. Fistula closure was observed in 62.5% of cases. Improvement of inflammatory and nutritional markers was noted in patients that responded to EEN. Subjects included in these two studies continued EEN continuing for 12 weeks. 

According to these reports the use of EEN in penetrating CD seems promising and warrant larger prospective studies. 

#### 6.5.2. Stricturing CD

The outcome of 12 weeks of EEN therapy in 59 patients with inflammatory strictures evaluated in a prospective cohort by Hu et al. [154]. In total 50 out of 59 (84.7%) subjects completed the EEN course, of which 81.4% of patients achieved symptomatic remission while 35 subjects achieved radiological (bowel wall thickness, luminal diameter, and luminal cross-sectional area) remission of the inflammatory strictures confirming the potential anti-inflammatory effect of EN. In particular, bowel wall thickness decreased by 59% with a 331% increase of luminal cross-sectional area at week 12. Clinical symptoms, nutritional markers and inflammatory markers improved during treatment with EEN. The nine patients that did not complete EEN underwent surgery for progressive bowel obstruction. 

The efficacy of EEN in CD patients with incomplete intestinal obstruction due to stenosis or stricturing at various locations (duodenal, jejunal, ileal, ileocolonic or colonic) was evaluated in another Chinese study by Xie et al. [155]. EEN was administered via a nasogastric or percutaneous endoscopic gastrostomy tube (with or without a jejunal extension) for a period of 12 weeks. Clinical and biochemical remission was achieved in 75% of individuals. Of note, during therapy, seven subjects of the cohort required surgery. 

Finally, a recent case report described the use of two weeks of EEN delivered continuously via a nasogastric tube in one patient presenting with stricturing duodenal CD on a background of ileal CD [156]. Both clinical and endoscopic improvement of the stricture were documented but not quantified. 

#### 6.5.3. Extra-Intestinal CD

Overall, the role of EEN in the management of extraintestinal manifestations of CD has not been fully determined. Anecdotal report exists on beneficial effect in erythema nodosum and IBD-associated joint symptoms. There is a single case report of oral-facial granulomatosis resolving after two days of commencing elemental diet, suggest that EN may be effective in CD patients with orofacial involvement [157]. Specifically, what the role of EN for either isolated oral-facial granulomatosis or oral-facial granulomatosis in conjunction with luminal disease is, requires further evidence. 

## 7. Role of EN in the Perioperative Setting

EN has been used prior to elective surgery to reduce surgical complications [158]. It has been reported to down-stage disease activity, the need for urgent surgery and reduce post-operative complications such as abscesses or anastomotic leak, by reducing inflammation, improving nutritional status and decreasing antigenic load through bowel rest [159]. 

Patients with CD who require elective or urgent surgery are at increased risk of malnutrition. As might be expected, the reasons for this are multiple and include malabsorption, nutrient loss, poor dietary intake and increased nutritional requirements driven by systemic inflammation [34]. The main goals of improving nutrition in the peri-operative period include downstaging disease activity and extent through the inflammation modulating effect of EN [43] and reducing post-operative morbidity, principally intra-abdominal septic complications such as leaks, abscesses, and enterocutaneous fistulae [158,159]. 

The most recent guidelines from ESPEN [34] recommend that surgery should be postponed for at least 7–14 days if possible, to allow for nutritional optimisation (EN is preferred over PN) in malnourished patients with CD. PN is recommended if >60% of energy needs cannot be met via the enteral route. Additional review articles similarly emphasise the need for pre-operative EN for surgical optimization [160,161]. An algorithm has been suggested by Grass et al. [149] to implement current guidelines on perioperative EN and PN. 

### 7.1. The Status of Pre-Operative Malnutrition in Patients With Crohn’s Disease

Although establishing the degree of malnutrition in CD patients can be difficult, recent guidelines by ESPEN have defined severe malnutrition in CD as an Albumin <30 g/L, BMI <18.5 kg/m^2^ and weight loss >10–15% within six months [34]. CD affecting the small bowel leads to protein-calorie and specific nutrient malnutrition. Thus, malnutrition is common in remission and active disease, is linked to malabsorption, and is more likely in hospitalised patients [162,163]. Furthermore, sarcopenia, a loss in lean muscle mass and strength, has been shown in subgroup analyses to be a predictor of surgical complications [164,165]. Data are limited but it is estimated that up to 85% of patients awaiting surgery are malnourished [166]. ESPEN, accordingly, recommends that patients with CD undergoing surgery are screened for malnutrition, as with for general surgery [34,167]. 

### 7.2. Enteral Nutrition Reduces 30-Day Post-Operative Intra-Abdominal Septic Complications

Whilst the trend in the rate of surgery in CD has improved over time, the 10-year risk of surgery remains at over 45% [168]. Within this patient population, the rate of important intra-abdominal septic complications of surgery may reach 50% with low albumin, pre-operative corticosteroids, pre-operative abscess and prior surgery among the key risk factors [169]. Five recent institutional retrospective cohort studies have reported on the effect of EN on rates of post-operative intra-abdominal septic complications of which four have been reviewed in detail elsewhere [170]. 

Li et al. [171] evaluated four weeks of EEN as semi-elemental feed delivered by a nasogastric tube in those awaiting surgery for non-perianal fistulising CD (enterocutaneous fistulae). For 30-day post-operative outcomes, they found that EN was a significant independent protective factor against intra-abdominal septic complications and non-infectious complications such as ileus and increased the immunosuppressant-free interval after surgery. Zhu et al. [172] similarly evaluated semi-elemental EEN at a median duration of six weeks within a cohort of patients with percutaneously undrainable abscesses co-treated with antibiotics. They showed a decreased risk of IASC and in the duration of post-operative hospitalisation in the EEN group. Ge et al. [173] compared semi-elemental EEN with general ONS in patients undergoing laparoscopic surgery finding reduced surgical site infections but not a significant difference in intra-abdominal septic complications. 

Heerasing et al. [174] assessed oral polymeric EEN of a mean duration of six weeks and also reported significantly reduced intra-abdominal septic complications in an EEN cohort who had lower BMI and higher disease activity at baseline compared to matched controls. Beaupel et al. [175] selected patients who were due elective ileocaecal resection if they had >10% weight loss, corticosteroids use or partial small bowel obstruction (defined as “high risk”). Following a median duration of three weeks of oral polymeric EEN, this cohort had no significant difference in intra-abdominal septic complications compared with the non-EEN “low risk” cohort, and 62.5% discontinued corticosteroids in the pre-operative period [175]. 

### 7.3. Enteral Nutrition Can Improve Inflammatory and Nutritional Indices

The effect of EEN on biochemical indices of inflammatory activity, such as c-reactive protein, erythrocyte sedimentation rate, and albumin, is well-established in the paediatric CD population [176]. A low albumin (<25 or 30 g/L) has been identified as a risk factor for intra-abdominal sepsis after surgery in CD [169,177,178]. Several studies of pre-operative optimisation with EEN demonstrate significant improvements in albumin, c-reactive protein and BMI [179,180], and c-reactive protein in isolation [174]. 

### 7.4. EN Can Reduce the Need for Surgery

Zhu et al. [172] also reported a significant reduction in the cumulative surgical rate of their EEN cohort and furthermore a total avoidance of surgery in 15% of that group. Of the 51 patients receiving EEN in the Heerasing et al. [174] cohort, 25% did not ultimately require surgery. 

### 7.5. EN Can Reduce Post-Operative Recurrence

Wang et al. [179] demonstrated a significant short-term reduction in endoscopic recurrence after bowel resection in patients who received four weeks of pre-operative EEN compared with controls in a retrospective study. Patient cohorts were comparable for phenotypic and therapeutic variables, nearly all undergoing surgery for fibrostenotic CD. At six months, a Rutgeerts score ≥i2 was lower in the EEN group compared with controls (*p* = 0.03), but non-significant at 12, 18 and 24 month endoscopic assessments. Clinical recurrence showed no significant difference between cohorts. 

Yamamoto et al. [101] assessed a low fat diet with supplemental overnight EN by enteral tube in a five-year prospective cohort study. This demonstrated a significant, sustained lower post-operative recurrence versus controls using the end-points of infliximab treatment (10% vs. 45%) and re-operation (5% vs. 25%). However, an important limitation to the generalisability of this data was the onerous demand of daily nasogastric tube self-intubation to administer EN in a cohort selected for their prior compliance compared with controls. 

### 7.6. Choice of Nutritional Support

No randomised controlled trials exist comparing elemental with polymeric EEN in pre-operative CD and the heterogeneity of existing retrospective cohort studies limits direct comparison. No direct comparison between EN and PN in pre-operative CD has been conducted with recommendations for EN over PN reflecting logistical and safety practicalities rather than clinical efficacy evidence [20]. 

## 8. Conclusions

EN is largely under-utilized in adult CD. A more robust evidence-base is required to elucidate the optimal use of EN in treatment algorithms. Healthcare professionals should consider offering EN as a potential first-line therapy in certain situations—particularly in motivated patients and those who prefer a non-pharmacological approach. 

There is still substantial variation in the use of EEN in different parts of the world both in adult and paediatric CD. EEN is not routinely used in North America [181] and only 4% of American gastroenterologists use it regularly to manage mild to moderately active paediatric CD, compared to 62% of Western European gastroenterologists [182]. EEN is widely used as primary induction therapy for children and adolescent CD in European countries and in New Zealand, Australia, Canada, and Asian countries [58,124]. 

The most recent Cochrane review suggests that EEN is superior to corticosteroids in paediatric CD, but slightly inferior in adult CD for the induction of remission [23]. The use of EN in CD should be considered in various clinical scenarios, including the perioperative setting. The data available on a per-protocol analysis do not support superiority of corticosteroids compared to EEN in induction therapy suggesting that adherence is the main factor affecting efficacy. Regardless, the use of EN deserves consideration for the induction of remission in CD due to its positive effects, such as mucosal healing and a lack of sustained and serious adverse events, compared to corticosteroids, such as osteoporosis. EN can be used safely and improves the therapeutic effect of other CD therapies, such as immunomodulators, available to treat CD [160]. Although the studies on the efficacy of EN in complicated CD are heterogeneous in terms of protocols of EEN therapy, they consistently show that EEN has a role in treating stricturing/penetrating disease and the extra-intestinal manifestation of CD. The longer-term outcomes in these settings are unknown. 

The routes of feeding do not seem to interfere with the efficacy of EN treatment and oral administration can replace invasive tube feeding to induce remission if adequate volumes can be consumed. However, palatability issues of elemental feed, as seen in pre-operative optimisation studies, often necessitates enteral tube delivery. It is the palatability in adults, contributing to a 41% pooled drop-out rate of EEN from seven induction of remission studies [17], that is an important consideration for widespread treatment in pre-operative and in the general CD populations. 

Finally, none of the studies available in the literature report data on the cost-effectiveness of EN [108]. Particularly, in the current climate of limited resources, a lack of health economic data for EN represents a crucial deficiency. 

To achieve clarity and the standardization of therapy, there is a need to address the uncertainty around the use of EN, particularly regarding the duration of treatment, whether a concurrent oral intake should be permitted, and how to reintroduce food once remission has been induced. There is a wide range of uptake of EN therapy worldwide. 

Physicians’ attitudes towards EN and patient compliance are the main barriers to the use of EN in the adult population. New formulations need to be developed to improve palatability and increase compliance if EN is to become a realistic treatment option. 

The development of palatable EN formulations that can be delivered without use of a nasogastric tube may lead to increased patient adherence with this therapy. However, in cases where there may be issues with gastrointestinal absorption and weight gain is required the nasogastric route may be preferential. 

As most studies on PEN use an elemental formulation via nasogastric tube, further research from a formal adequately powered equivalence trial is required to determine the optimal EN formulation for use in people with quiescent CD [35]. 

Our review provides a comprehensive update of the available literature regarding the role of EN in CD. It offers a useful reference tool to guide clinical decision-making, and support EN use in adult CD to improve clinical, surgical and nutritional outcomes, and to reduce concurrent side effects. 

## Figures and Tables

**Table 1 nutrients-11-02222-t001:** Most commonly used UK based nutritionally complete oral enteral feeds used in adult CD.

Type of Formulation	Name	Company	Presentation	Fat Profile [Long Chain Triglycerides (LCT)/MCT: %/%]	Calories/Protein Per 100 mL	Osmolality (Mosm/Kg)
Elemental	E028 extra powder	Nutricia	100 g Sachet	52/48	89/2.5	340 *
	E028 extra liquid	Nutricia	250 mL carton	65/35	86/2.5	725 **
	Emsogen	Nutricia	100 g Sachet	2.4/97.5	66/1.9	290
Semi-elemental	Vital 1.5	Abbott	200 mL Bottle	-/45	150/6.75	630
	Peptamen 1.5	Nestle	250 mL Bottle	-/72	150/6.8	check
Polymeric	Ensure plus milkshake style	Abbott	200 mL Bottle	-/0	150/6.25	660
	Fortisip	Nutricia	200 mL Bottle	-/-	150/6	455
	Fresubin Energy	Fresenius Kabi	200 mL Bottle	19/-	150/6 *	500
	Modulen IBD	Nestle	400 g tin	57/25	100/3.6 ***	340/539 ****

* unflavoured; ** orange and pineapple; *** This can be concentrated to 1.5 kcal/mL (at this concentration 150/5.4); **** at 1.5 kcal/mL concentration; ‘-’ No constituent figures available from manufacturers.

**Table 2 nutrients-11-02222-t002:** Guidelines currently in use for enteral nutrition in Crohn’s disease. The main points are described.

Guideline	Main Points
ESPEN 2017 [34]	A multi-disciplinary guideline based on extensive systematic review of the literature, with expert opinion input, producing 64 recommendations. Diets with a high fruit and vegetable content, rich in n-3 fatty acids, and relatively deplete of n-6 fatty acids are associated with a lower risk of developing CD. While protein requirements are elevated in active IBD (therefore necessitating increased intake relative to the general population), protein needs in remission are more commonly normal. Iron supplementation is advised in all IBD patients with iron deficiency anaemia, with oral iron recommended as the first-line treatment in patients with mild anaemia and clinically inactive disease while intravenous iron is advised in patients with clinically active IBD, with haemoglobin below 100 g/L, and in those requiring erythropoiesis-stimulating agents. In CD patients with both intestinal strictures (or stenosis) and obstructive symptoms, a diet with adapted texture, or post-stenosis enteral nutrition is advised. In patients with active disease and those who are steroid-treated, serum calcium and vitamin D levels should be monitored and supplemented as necessary to maintain bone mineral density. Exclusion diets are not recommended to achieve remission in active CD, even if the patient suffers from individual intolerances. Probiotics should not be used for treatment of active CD. ONSs are advised as a first line dietary treatment when artificial nutrition is indicated, albeit a minor supportive strategy in addition to normal food. If oral feeding is insufficient then tube feeding should be considered using formulae or liquids, as a preference over parenteral feeding. In children and adolescents with acute active CD, EEN is recommended as first line treatment to induce remission. Probiotic therapy should not be used for maintenance of remission in CD. All IBD patients in remission should undergo counselling by a dietician as part of the multidisciplinary approach to enhance nutritional therapy and avoid malnutrition and secondary disorders. In patients who are pregnant, iron status and folate levels should be monitored regularly with deficiencies of iron/folic acid/vitamin B9 treated with supplementation accordingly.
Cochrane Systematic Review 2019 [119]	A retrospective systematic review, including of 18 randomized controlled trials with 1878 patients, assessing the impacts of different dietary strategies on both active and inactive CD. At four weeks, 100% of patients in the high fibre and low refined carbohydrates diet group experienced clinical remission of active CD compared to 0% in the control group. 50% of participants in the symptoms-guided diet group achieved clinical remission of active CD while 0% of the control group participants achieved remission. There is insufficient evidence to support the role of exclusion diets in clinical remission rates of active CD. There is insufficient evidence to support the role of exclusion diets in reducing clinical relapse rates for inactive CD. More thorough evaluation of the benefits and harms of specific compositions of dietary interventions is required through further randomized controlled trials.
European Crohn’s and Colitis Organisation (ECCO)/European Society for Paediatric Gastroenterology Hepatology and Nutrition(ESPGHAN) 2014 [120]	A set of evidenced based and consensus driven guidelines for paediatric-onset CD formulated by an expert panel of 33 IBD specialists to optimise treatment of children and adolescents with CD, with bespoke management plans based on benefit-risk analyses according to different clinical contexts. EEN is recommended as a first line therapy to induce remission in children with active luminal CD. PEN should not be used for induction of remission. The use of EEN as induction therapy usually runs for 6–8 weeks, with an absence of clinical response after two weeks considered to warrant consideration for alternative treatment. EEN is more ideally implicated for children with poor growth, a low BMI and those with a catabolic state (e.g., hypoalbuminemia). In cases where oral EEN is not tolerated, a nasogastric tube may be considered. However, the psychological and monetary impact of this option requires cautious consideration in regard to alternative strategies, such as a limited course of steroids treatment.
Croatian Guidelines 2010 [121]	A set of guidelines governing the use of enteral nutrition in CD, developed by an interdisciplinary expert group of IBD-related clinicians based on relevant evidence-based literature and experience. Enteral nutritional therapy is an effective first-line treatment for paediatric patients with active CD. There is no difference in the efficacy of elemental, oligomeric and polymeric enteral formulae. Therefore, polymeric formulae are advised on account of its more acceptable palatability, and relatively lower cost. There is evidence of beneficial effects supporting the use of novel nutritional therapeutically-enriched feeds, such as TGF β2 enteral feeding.
North American Guidelines 2012 [122]	An ‘aid’ review of the use of enteral nutrition as paediatric CD therapy by the Enteral Nutrition Working Group, composed of six clinicians with expertise and/or experience in exclusive enteral nutrition. EEN provides an alternative to corticosteroids in the induction of clinical remission in paediatric CD and should be used as a first-line treatment, irrespective of active disease location. A treatment course of at least eight weeks of EEN is advised for induction, although benefits may be yielded for up to 12 weeks in some cases. In children with stunted growth or pubertal delay, continued PEN, alongside additional treatment interventions may yield clinical improvement. Polymeric formulae are cheaper, more palatable, and are associated with better weight gain when compared with an elemental diet. They are thus more conducive to oral administration in the paediatric population. Multidisciplinary coordination between nursing staff, medical clinicians, and dietetic services are more likely to enhance the impact of an EEN programme, with evidence also supporting the roles of social workers and psychology support staff where appropriate.
Japanese Guidelines 2005–2006 [56,123]	Japanese guidelines on the management of CD recommend nutritional therapy as both a first-line treatment for remission and as a maintenance therapy thereafter. Japanese patients with CD have a proven lower mortality rate because of this, in comparison to those who do not receive enteral feeding. For home-based tube enteral feeding, administration of nutrition overnight is advised. Excessive amounts of long-chained fatty acids in the composition of feed can hamper the beneficial effects of enteral nutrition. This adversity can be bypassed through the use of medium-chained triglycerides.

**Table 3 nutrients-11-02222-t003:** Examples of studies that examined remission of CD with EN.

Study	Design	Main Results
Giaffer et al. [49]	Randomized trial, not double blind 30 participants: 8 males, 22 females 16 participants assigned elemental diet and 14 assigned the polymeric diet	Remission rate determined by Crohn’s Disease Activity Index (CDAI) <150 after 10 days (Elemental 12/16 versus non-elemental 5/14, RR = 2.10, *p* > 0.05)
Grogan et al. [38]	Randomized trial, double blind 41 paediatric patients with newly-diagnosed CD and no isolated colonic disease 20 patients received elemental diet and 21 received the polymeric diet	Remission rate determined by Pediatric Crohn’s Disease Activity Index (PCDAI) <11 at 6 weeks (Elemental 14/20 versus non-elemental 15/21, RR = 0.98, *p* > 0.05)
Kobayashi et al. [39]	Randomized trial, not double blind 19 participants 10 participants assigned elemental diet and 9 assigned the polymeric diet	Remission rate determined by CDAI <150 after 24 days (Elemental 7/12 versus non-elemental 6/10, RR = 0.97, *p* > 0.05)
Mansfield et al. [44]	Randomized trial, not double blind 44 participants: 16 males, 28 females 22 participants assigned elemental diet and 22 assigned the semi-elemental diet	Remission rate determined by CDAI reduction of 100 points or 40% of the initial value after 28 days (Elemental 8/22 versus non-elemental 8/22, RR = 1.00, *p* > 0.05)
Middleton et al. [45]	Randomized trial, not double blind 76 participants 58 participants assigned elemental diets and 18 assigned the semi-elemental diet	Remission rate determined by Harvey-Bradshaw Index <3 after 21 days (Elemental 36/58 versus non-elemental 13/18, RR = 0.86, *p* > 0.05)
Park et al. [40,131,132]	Randomized trial, double blind 14 participants: 1 male, 13 females 7 participants assigned elemental diet and 7 assigned the polymeric diet	Remission rate determined by Harvey-Bradshaw Index <2 after 28 days (Elemental 2/7 versus non-elemental 5/7, RR = 0.40, *p* > 0.05)
Raouf et al. [41]	Randomized trial, not double blind 24 participants 13 participants assigned elemental diet and 11 assigned the polymeric diet	Remission rate determined by Harvey-Bradshaw Index <4 after 21 days (Elemental 9/13 versus non-elemental 8/11, RR = 0.95, *p* > 0.05)
Rigaud et al. [42]	Randomized trial, not double blind 30 participants: 18 males, 12 females 15 participants assigned elemental diet and 15 assigned the polymeric diet	Remission rate determined by CDAI <150 after 28 days (Elemental 10/15 versus non-elemental 11/15, RR = 0.91, *p* > 0.05)
Royall et al. [46]	Randomized trial, double blind 40 participants: 23 males, 17 females 19 participants assigned elemental diet and 21 assigned the semi-elemental diet	Remission rate determined by CDAI <150 after 21 days (Elemental 16/19 versus non-elemental 15/21, RR = 1.18, *p* > 0.05)
Sakurai et al. [47]	Randomized trial, not double blind 36 participants: 30 males, 6 females 18 participants assigned elemental diet and 18 assigned the non-elemental diet	Remission rate determined by CDAI decrease of at least 40% or by 100 or more after 42 days (Elemental 12/18 versus non-elemental 13/18, RR = 0.92, *p* > 0.05)
Verma et al. [43]	Randomized trial, double blind 21 participants: 8 males, 13 females 10 participants assigned elemental diet and 11 assigned the polymeric diet	Remission rate determined by CDAI <150 or CDAI decrease of 100 points from baseline level after 28 days (Elemental 8/10 versus non-elemental 6/11, RR = 1.47, *p* > 0.05)

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
