# Peer review of "Enteral Nutrition in Adult Crohn’s Disease: Toward a Paradigm Shift"

_nutrients, 2019, doi:10.3390/nu11092222_

Round 1

Reviewer 1 Report

This review comprises a compendium on the use of EN in Chron's disease, with updated and critically reviewed data. The information is well organized, the objectives were clearly defined and the results presented are consistent with them. Also, a good bibliographic exploration is appreciated.
However, there are some aspects of form that need to be corrected in order to improve the presentation, which are detailed in the attached document.

Author Response

Review and responses

Line 19: "contains" should be changed to ",containing" (with a comma before containing).

Response: we have now addressed this comment and text has been replaced.

Line 20: Add a hyphen between low and risk (low-risk), as well as the conjunction "and" between low-risk/minimally invasive therapy (low-risk and minimally invasive therapy).

Response: we have now addressed this comment and text has been replaced.

Line 22: Western should be capitalized.

Response: we have now addressed this comment and text has been replaced.

Line 62: Western should be capitalized.

Response: we have now addressed this comment and text has been replaced.

Line 91: Enter page break so that the subtitle is not at the end of the page.

Response: we have now addressed this comment

Table 1: Please correct the fat profile column (poor visualization of the column heading).

Response: we have now addressed this comment and allowed text be visualised.

Table 1: Esmogen has no information related to the exact content of each sachet. Please include this information.

Response: we have now addressed this comment and added missing information.

Line 125: Replace “used in CD” with “used for CD”.

Response: we have now addressed this comment and text has been replaced.

Line 162: Replace “sufflower” with “sunflower”.

Response: we have now addressed this comment and text has been replaced.

Line 174: Delete the first into in “or into inserted into the jejunum”.

Response: we have now addressed this comment and text has been replaced.

Line 188: Add the preposition “the” before adequate and also before route in the same line.

Response: we have now addressed this comment and text has been replaced.

Line 191: Add the preposition “a” before standard practice in Japan…

Response: we have now addressed this comment and text has been replaced.

Line 195: Society should be capitalized.

Response: we have now addressed this comment and text has been replaced.

Line 237: Replace “microflora” for microbiota.

Response: we have now addressed this comment and text has been replaced.

Line 241: Replace “microflora” for microbiota.

Response: we have now addressed this comment and text has been replaced.

Line 292: Add “is” before contraindicated.

Response: we have now addressed this comment and text has been replaced.

Table 2:

- In Cochrane Systematic Review 2019 section, in the second point add a space after 50%.

Response: we have now addressed this comment and text has been replaced.

- In European Chron’s and Colitis Organisation section, to homogenize the style of writing, replace “pediatric-onset” with paediatric-onset.

Response: we have now addressed this comment and text has been replaced.

- In Croatian guidelines 2010 section, in the first point replace pediatric with paediatric.

Response: we have now addressed this comment and text has been replaced.

Line 351: Add a period after vs.

Response: we have now addressed this comment and text has been replaced.

Line 391: Replace “is” with “are” after foods.

Response: we have now addressed this comment and text has been replaced.

Line 395: Change “versus” for vs.

Response: we have now addressed this comment and text has been replaced.

Line 459: Delete the period after EN.

Response: we have now addressed this comment and text has been replaced.

Line 478: Replace “102” with “One hundred and two”.

Response: we have now addressed this comment and text has been replaced.

Line 485: Add a space after the parenthesis.

Response: we have now addressed this comment and text has been replaced.

Line 696: Add an endpoint after “worldwide”.

Response: we have now addressed this comment and text has been replaced.

Line 704: Delete the “n” between via and nasogastric tuve.

Response: we have now addressed this comment and text has been replaced.

Reviewer 2 Report

I would like to congratulate the authors on a thorough review of this topic. the following suggestions are aimed at improving the clarity and quality of the final published version.

Line 83 – Update to “A search of literature…” or “A literature search…” Table 1 – check heading text as some elements are not fully legible. Line 113 – “as the protein source” is perhaps not strictly correct, as amino acids are not proteins. Would “to meet protein requirements” or similar be more correct? Line 167 – I think that the word “not” is missing before “any”. Line 214 – the statement in this sentence is too vague. Are the authors suggesting that there appear to be no other similar studies in this area? Beyond reproduction of the approach of Rubio et al, are any other studies/research questions warranted? Line 215-219 – I think this component needs a bit more discussion in relation to earlier points. Impacts on appetite are generally assessed as acute responses to a single meal, which can’t really be extrapolated to long-term (lean) weight gain or reduction of weight loss. As the authors have already noted, the issue of compliance means that many individuals may simply not be able to adhere to enteral nutrition approaches long-term. I would presume that adherence of a few weeks would be necessary to see a measurable improvement in body weight status let alone a biologically significant one. Line 237-240 – this section has been written as a series of facts. I suggest that the authors reconsider this paragraph to align with the more appropriate, scientifically conservative style of presentation they have used elsewhere. “There appears to be/may be [an increase/decrease…]” seems more appropriate based on the evidence presented. Section 4.2 – case-control studies have often suggested a difference between patient groups and asymptomatic individuals in terms of the gut microbiota. It’s currently uncertain whether this observation is cause or effect and the use of term “dysbiosis” suggests a movement away from an “ideal” or “target” microbiota that has not been (widely) accepted or confirmed as a unambiguous marker of host health. I believe that the authors are correct to raise the changes in phyla and diversity that hints as microbial changes that could be construed as less than ideal but a) need to provide appropriate background information for the wider audience to understand this concept and b) be careful to avoid the suggestion of causality with disease progression and a linearity in Line 286 – as “is” before “used”. Line 303 – update to “The main points…”. Line 394 – update to “The latest studies…” Line 408 – I suggest updating the title to “Examples of studies…” or “Major studies…” or similar. Main/The main does not seem to be quite correct. Line 508 – I’m not sure the country of authorship is particularly relevant to the point being made here. I suggest updating to “The authors of both studies did not provide…” or similar. Section 6.5 – in each case, the authors have provided a worthwhile oversight of limited research into complicated manifestations of CD. I suggest that the authors consider including some discussion in relation to (the limited but generally positive) evidence of potential benefit in relation to the risk of negative effects. I imagine that these would compare favourably to other available approaches (which presumably may have limited evidence bases). Line 567-568 – in relation to the point above, how does 7 requiring surgery relate to outcomes with other treatment options? Bear in mind that the readership of Nutrients is broad and most will not have a clinical background. Line 624 – update to “of a mean duration of 6 weeks or similar”.

Author Response

Revision and responses

Line 83 – Update to “A search of literature…” or “A literature search…”

Response: we have now addressed this comment and text has been replaced.

Table 1 – check heading text as some elements are not fully legible.

Response: we have now addressed this comment and text has been replaced.

Line 113 – “as the protein source” is perhaps not strictly correct, as amino acids are not proteins. Would “to meet protein requirements” or similar be more correct?

Response: we have now addressed this comment and text has been replaced.

Line 167 – I think that the word “not” is missing before “any”.

Response: we have now addressed this comment and text has been replaced.

Line 214 – the statement in this sentence is too vague. Are the authors suggesting that there appear to be no other similar studies in this area? Beyond reproduction of the approach of Rubio et al, are any other studies/research questions warranted?

Response: we thank the reviewer for this comment. There are limited studies in this area with Rubio et al. being the main one discussed. No more questions are warranted for this section but better designed studies comparing oral vs nasogastric feeding for delivery of enteral nutrition for inducing remission in Crohn’s disease. A clarifying sentence has been added.

Line 215-219 – I think this component needs a bit more discussion in relation to earlier points. Impacts on appetite are generally assessed as acute responses to a single meal, which can’t really be extrapolated to long-term (lean) weight gain or reduction of weight loss. As the authors have already noted, the issue of compliance means that many individuals may simply not be able to adhere to enteral nutrition approaches long-term. I would presume that adherence of a few weeks would be necessary to see a measurable improvement in body weight status let alone a biologically significant one.

Response: we thank the reviewer for this comment. We agree with the reviewer that this is not possibly to extrapolate long term and have added a necessary sentence to explain this.

Line 237-240 – this section has been written as a series of facts. I suggest that the authors reconsider this paragraph to align with the more appropriate, scientifically conservative style of presentation they have used elsewhere. “There appears to be/may be [an increase/decrease…]” seems more appropriate based on the evidence presented.

Section 4.2 – case-control studies have often suggested a difference between patient groups and asymptomatic individuals in terms of the gut microbiota. It’s currently uncertain whether this observation is cause or effect and the use of term “dysbiosis” suggests a movement away from an “ideal” or “target” microbiota that has not been (widely) accepted or confirmed as a unambiguous marker of host health. I believe that the authors are correct to raise the changes in phyla and diversity that hints as microbial changes that could be construed as less than ideal but a) need to provide appropriate background information for the wider audience to understand this concept and b) be careful to avoid the suggestion of causality with disease progression and a linearity in Line 286 – as “is” before “used”.

Response: We thank the reviewer for this comment. We have now addressed this comment and modified wordings and explained the possibility of dysbiosis being a consequence and not cause with appropriate references.

Line 303 – update to “The main points…”.

Response: we have now addressed this comment and text has been replaced.

Line 394 – update to “The latest studies…”

Response: we have now addressed this comment and text has been replaced.

Line 408 – I suggest updating the title to “Examples of studies…” or “Major studies…” or similar. Main/The main does not seem to be quite correct.

Response: we have now addressed this comment and text has been replaced.

Line 508 – I’m not sure the country of authorship is particularly relevant to the point being made here. I suggest updating to “The authors of both studies did not provide…” or similar.

Response: we have now addressed this comment and text has been replaced.

Section 6.5 – in each case, the authors have provided a worthwhile oversight of limited research into complicated manifestations of CD. I suggest that the authors consider including some discussion in relation to (the limited but generally positive) evidence of potential benefit in relation to the risk of negative effects. I imagine that these would compare favourably to other available approaches (which presumably may have limited evidence bases).

Response: we have now addressed this comment. We have added a text explaining enteral nutrition versus parenteral nutrition and that EN is overall better in terms of risk/benefit compared to parenteral nutrition.

Line 567-568 – in relation to the point above, how does 7 requiring surgery relate to outcomes with other treatment options? Bear in mind that the readership of Nutrients is broad and most will not have a clinical background.

Response: We thank the reviewer for this comment. In this setting, we are describing the use of enteral in the perioperative period as a nutritional mean. It is not being descried against surgery as to the effectiveness of treating Crohn’s. To that end sometimes enteral nutrition has sometimes delayed or postponed surgery because it has induced remission with success, hence the patient has not needed surgery any more. WE hope his clarifies this section.

Line 624 – update to “of a mean duration of 6 weeks or similar”

Response: we have now addressed this comment and text has been replaced.